# SARS-CoV-2 Evolution among Oncological Population: In-Depth Virological Analysis of a Clinical Cohort

**DOI:** 10.3390/microorganisms9102145

**Published:** 2021-10-14

**Authors:** Florian Laubscher, Samuel Cordey, Alex Friedlaender, Cecilia Schweblin, Sarah Noetzlin, Pierre-François Simand, Natacha Bordry, Filipe De Sousa, Fiona Pigny, Stephanie Baggio, Laurent Getaz, Pierre-Yves Dietrich, Laurent Kaiser, Diem-Lan Vu

**Affiliations:** 1Laboratory of Virology, Division of Laboratory Medicine, Geneva University Hospitals, 1205 Geneva, Switzerland; Florian.Laubscher@hcuge.ch (F.L.); Samuel.Cordey@hcuge.ch (S.C.); cecilia.schweblin@hcuge.ch (C.S.); fiona.pigny@hcuge.ch (F.P.); Laurent.kaiser@hcuge.ch (L.K.); 2Faculty of Medicine, University of Geneva, 1206 Geneva, Switzerland; stephanie.baggio@hcuge.ch (S.B.); Laurent.Getaz@hcuge.ch (L.G.); Pierre-Yves.Dietrich@hcuge.ch (P.-Y.D.); 3Department of Oncology, Geneva University Hospitals, 1205 Geneva, Switzerland; alex.friedlaender@hcuge.ch (A.F.); sarah.noetzlin@hcuge.ch (S.N.); pierre-francois.simand@hcuge.ch (P.-F.S.); natacha.bordry@hcuge.ch (N.B.); Filipe.FernandesdeSousa@hcuge.ch (F.D.S.); 4Division of Prison Health, Geneva University Hospitals, 1205 Geneva, Switzerland; 5Division of Infectious Diseases, Geneva University Hospitals, 1205 Geneva, Switzerland; 6Center for Emerging Viruses, Geneva University Hospitals, 1205 Geneva, Switzerland

**Keywords:** SARS-CoV-2, minority variants, high-throughput sequencing, oncological patients, compartment

## Abstract

Background: Oncological patients have a higher risk of prolonged SARS-CoV-2 shedding, which, in turn, can lead to evolutionary mutations and emergence of novel viral variants. The aim of this study was to analyze biological samples of a cohort of oncological patients by deep sequencing to detect any significant viral mutations. Methods: High-throughput sequencing was performed on selected samples from a SARS-CoV-2-positive oncological patient cohort. Analysis of variants and minority variants was performed using a validated bioinformatics pipeline. Results: Among 54 oncological patients, we analyzed 12 samples of 6 patients, either serial nasopharyngeal swab samples or samples from the upper and lower respiratory tracts, by high-throughput sequencing. We identified amino acid changes D614G and P4715L as well as mutations at nucleotide positions 241 and 3037 in all samples. There were no other significant mutations, but we observed intra-host evolution in some minority variants, mainly in the ORF1ab gene. There was no significant mutation identified in the spike region and no minority variants common to several hosts. Conclusions: There was no major and rapid evolution of viral strains in this oncological patient cohort, but there was minority variant evolution, reflecting a dynamic pattern of quasi-species replication.

## 1. Introduction

Immunocompromised patients present the risk of a longer SARS-CoV-2 shedding duration compared to immunocompetent ones. This has led the Center for Disease Control to distinguish between non-immunocompromised and immunocompromised cases in their symptom-based strategy for discontinuing transmission-based precautions [1]. Several case reports identified SARS-CoV-2 RNA persistence lasting up to 5 months, associated with virus genetic evolution in specifically highly immunocompromised patients [2,3]. Viable virus was detected more than 2 months after infection in such patients [4,5]. In a specific cohort of renal transplant recipients, 5 among 42 patients had upper respiratory tract viral loads > 3 log10 copies per reaction after 30 days from diagnosis [6].

These studies describe very specific situations in particularly highly immunocompromised patients. Yet, most oncological patients have an attenuated form of immunosuppression, either due to the disease itself, or due to immunosuppressive treatment, and little is known about the shedding and viral evolution among this population that can be seen in any general practice consultation. The worse outcome observed during SARS-CoV-2 infection among oncological patients [7] could rely on the concept of “inflamm-aging”, where infection occurs in an already inflamed environment, favoring an inappropriate innate response, a delayed adaptive response and dysregulated pro-inflammatory cytokine production [8]. Yet, the pathophysiology of SARS-CoV-2 infection in oncological patients can be complex, knowing, for example, that the SARS-CoV-2-associated functional exhaustion of cytotoxic T lymphocytes and natural killer cells can be reverted by checkpoint inhibitors. Thus, the outcome of oncological patients infected by SARS-CoV-2 could paradoxically be favored by such treatments. On the other hand, steroids, anti-IL6r and JAK inhibitors, which are treatments used for severe COVID, decrease the inflammatory response but may also prolong viral clearance [9,10]. Yet, although the virological mechanism of prolonged SARS-CoV-2 shedding is still debated [11], the longer the shedding, the higher the risk of emergence of intra-host mutations. Prolonged shedding in the upper respiratory tract of at-risk patients can be an issue for public health, but the emergence of variants from these patients can be all the more problematic.

The aim of this study was to describe a cohort of patients with oncological disease and SARS-CoV-2 infection and to identify the viral evolution among selected patients.

## 2. Materials and Methods

### 2.1. Setting, Study Population and Design

This observational retrospective study was conducted at the Geneva University Hospital (HUG), Switzerland. We collected clinical and laboratory data of adult patients with a known oncological disease and a diagnosis of SARS-CoV-2 infection confirmed by RT-PCR assay between February and June 2020. We then selected patients with at least one of the following investigations: RT-PCR for SARS-CoV-2 performed on a blood sample, either prescribed by the physician or performed for the purpose of this study, or on a bronchoalveolar lavage (BAL), and patients with several nasopharyngeal swabs (NPS) tested for SARS-CoV-2 by RT-PCR. All included patients provided written consent before enrollment. The study protocol was approved by the Geneva Cantonal Ethics Commission (project 2020-00931).

### 2.2. Real-Time Reverse-Transcription Polymerase Chain Reaction (RT-PCR) Assay and Unbiased High-Throughput Sequencing (HTS) Analysis

RT-PCR was performed on NPS using three commercial diagnostic methods: the Cobas 6800 SARS-CoV-2 RT-PCR (Roche, Rotkreuz, Switzerland), the BD SARS-CoV-2 reagent kit for BD Max system (Becton, Dickinson and Co., Franklin Lakes, NJ, US) and the Xpert^®^ Xpress SARS-CoV-2 assay (Cepheid, Sunnyvale, CA, US), according to the manufacturers’ instructions. For blood and BAL samples, the in-house Charité rtRT-PCR protocol [12] was used on RNA previously extracted using the NucliSens easyMAG extraction kit (bioMérieux, Marcy-l’Étoile, France). For all PCR assays, the positivity threshold was Ct values ≤40 for at least one target.

HTS analysis was performed using the RNA protocol adapted from a study previously published by Petty et al. [13]. Briefly, for each specimen, 220 μL was centrifuged at 10,000× *g* for 10 min to remove cells. Then, 200 μL of cell-free supernatant was treated with 40U of Turbo DNAse (Ambion, Rotkreuz, Switzerland), according to the manufacturer’s instructions. Nucleic acids were extracted with TRIzol (Invitrogen, Waltham, MA, US). Ribosomal RNA was removed using the Ribo-Zero Gold depletion kit (Illumina, San Diego, CA, US). Libraries were generated using the TruSeq total RNA preparation protocol (Illumina) with dual indexing. Library concentrations and sizes were analyzed using the Qubit (Life Technologies, Carlsbad, CA, US) and the 2200 TapeStation instruments (Agilent, Santa Clara, CA, US), respectively. Thereafter, libraries were loaded on the HiSeq 4000 platform (Illumina, San Diego, CA, US) using the 2 × 100 bp protocol with dual indexing. Duplicate reads were removed using cd-hit (v4.6.8). Low-quality and adapter sequences were trimmed out using Trimmomatic (v0.33). Reads were then mapped against the reference sequence MN908947 using snap-aligner (v1.0beta.18). Consensus for sequences with at least 10-fold coverage was then generated using a custom script. Minority variants were checked for nucleotide positions with a minimum of 20× coverage (script available at https://github.com/V-HTS/HTS, accessed on 25 June 2021). For each patient, only minority variants that reached a minimum of 20% in at least one sample were further considered and reported in the other samples from the same patient.

## 3. Results 

### 3.1. Patients’ Demographics and Clinical Presentation

We included a total of 54 patients with an oncological disease, including 35 patients with solid tumors and 20 with hematological cancers. Half of the patients had an active oncological treatment within the previous 3 months of infection (Appendix A). The median delay from symptom onset to diagnosis by RT-PCR was 3 days (IQR 6), and the main symptoms were fever, cough, shortness of breath and rhinorrhea. About 80% of patients were hospitalized for a median duration of 2 weeks. There were very few extra-pulmonary manifestations and bacterial co-infections, and the mortality rate was 22% (Appendix A).

### 3.2. RT-PCR Assay Screening

Sixteen patients had more than one NPS performed, and five (31.2%) had more than one positive NPS. The NPS was positive over a median of 28 days (range 15–114). Among the total cohort, four patients had one or more BAL during the course of SARS-CoV-2 infection, and all were positive at a median delay of 13 days (range 7–22) after the NPS with a median CT value of 24 (range 19.8–25). Viremia was performed among 21 patients at a median delay of 9 days after NPS (range 0–51). Eight (38.1%) were positive at a median delay of 8 days (range 0–25) from NPS with a median CT value of 35.2 (range 32.4–39.3) (Table 1).

### 3.3. HTS Investigations

Six patients were analyzed by unbiased HTS: three patients (i.e., #1, #2 and #6) with several NPS performed to investigate any significant intra-host temporal variability, and three others (#3, #4 and #5) with different types of samples available to investigate any intra-host compartmental variability (Table 2, Figure 1). Among the six analyzed patients, two received remdesivir (#2 and #5) and none received passive immunotherapy. All samples had the D614G mutation in the spike region, corresponding to the variant circulating in Europe during the first period of the pandemic. Similarly, compared to the reference sequence MN908947, all samples had nucleotide mutations at positions 241 (untranslated region) and 3037 (ORF1ab region) and the amino acid change P4715L in the ORF1ab region (nucleotide position 14408). 

Due to the insufficient genome and/or depth coverage, samples 1b, 2c, 5b, 5d and 5e are not reported in Figure 1.

Among the samples analyzed for temporal variability, we found no genetic differences in patient #2 between both strains collected at a 3-week interval. In contrast, in patient #6, one amino acid change was observed at the amino acid position 4136 in ORF1ab in the first two NPS samples collected (4136A in samples 6a and 6b) (Figure 1). Interestingly, the analysis of minority variants revealed the presence of the wild-type 4136V in samples 6a and 6a at 15% and 38.97%, respectively, which finally became predominant at 100% in 6c. Furthermore, our analysis revealed only two other minority variants among the three patients analyzed for the intra-host temporal variability: one observed in the two NPS collected for patient #2 at the nucleotide position 17010 (ORF1ab), and one in patient #6′s 6a and 6b samples at the amino acid position 6513 (R6513L) in ORF1ab. 

Neither temporal genetic differences nor minority variants were observed in the spike region. Although one nucleotide change was observed at position 21974 in sample 6b compared to the reference sequence MN908947, this genome region is not covered in samples 6a and 6c, making it impossible to determine whether this nucleotide change was already present in sample 6a or acquired subsequently. Of note, this is the only mutation found within the S gene.

Our unbiased HTS investigation did not reveal any intra-host compartmental variability mutations (Figure 1). Furthermore, only three minority variants were identified: none in patient #3, two in patient #4 at the nucleotide positions 15324 (ORF1ab region) and 29685 (untranslated region) in both the upper and lower respiratory samples (i.e., 4a and 4b, respectively) and one in patient #5 at the nucleotide position 26333 (T30I in the E gene) in both the upper and lower respiratory samples. 

No genetic differences or minority variants were observed in the spike region for any sample, but there was a difference in the proportion of minority variants recovered among samples within the same host. Appendix A summarizes the position of minority variants and their respective proportions for each sample. 

## 4. Discussion

The present study described the clinical characteristics and outcomes of a single-center cohort of oncological patients, mostly with solid tumors, and thus the results cannot be extrapolated to other populations. We also provided virological investigations performed during routine clinical care. A total of 80% of patients were hospitalized, 13% were intubated and the mortality rate was 22%. Five patients had more than one positive NPS over up to 114 days. Of note, prolonged shedding has been described even in immunocompetent patients with viable virus that can be seen in samples analyzed up to 100 days after symptom onset [14]. Interestingly, viremia retroactively performed on blood samples revealed that 40% of patients tested were viremic but at a low level. Several reports have described patients treated with rituximab, with prolonged and increasing SARS-CoV-2 viremia up to 26 days after infection [15,16]. Benotmane et al. even found an association between plasma viral load and disease severity and mortality in kidney transplant recipients [17].

Compared with the Wuhan reference strain, all of the samples sequenced in the present study had a single-nucleotide variation (SNV) at positions 241, 3037, 14408 and 23403, the last one corresponding to the D614G amino acid mutation on the S gene. These four SNVs were present in the majority of circulating strains during spring 2020 and were equally identified in the majority of consensus sequences in other studies [18,19]. Our HTS analyses did not reveal the presence of evolutionary mutations between two strains within the same host, searching for either intra-host temporal variability or intra-host compartmental variability. Similar to Aydillo et al., we found no mutations despite the fact that samples were collected several weeks apart [5]. Li et al. also sequenced two strains, resulting in no single-nucleotide mutation for one and one single non-synonymous change for the other [14]. On the other hand, Sepulcri et al. identified an in-host mutation [16]. These apparent discordant findings may rely on the random nature of point mutations that emerge at a rate of 1–2 nucleotides per month [20], coupled with the limitations of next-generation sequencing techniques that should be perfected to cover whole genomes with a sufficient depth coverage, and to identify emerging minor variants. 

In addition to looking for accumulating virus mutations in hosts with prolonged shedding, we analyzed the presence and evolution of minority variants within the same hosts. Although there were no minority variants common to several hosts, we identified intra-host evolution in some minority variants, with a proportion that varied from <1% to 40% of minority variants found in distinct samples within the same host. In patient #6, the first two samples retrieved 10 days apart had amino acid mutation 4136A as a consensus strain, but the last sample retrieved 3 weeks after the first one showed the wild-type 4136V as a consensus strain. Yet, 4136V was already found as minority variants in samples 6a and 6b, at 15% and 38%, respectively. This minority variant thus increased in proportion with time and finally became predominant. It is considered that minority variants under specific selective pressure may become predominant and provide a fitness advantage, ultimately influencing the epidemic, as seems to have occurred with the D614G mutation [21]. As mentioned earlier, prolonged shedding has been described even in immunocompetent patients, and similarly, it is also possible that our findings of minority variants are not specific to the oncological population but could also be found in “healthy” infected individuals.

The lungs could be the site of virus persistence [14]. Our findings show the absence of intra-host genetic differences in the consensus strains recovered from the upper and lower respiratory tracts in three hosts, which has potential implications as lower respiratory tract sampling is technically complex and involves higher risks both for patients and healthcare professionals. Knowing that both strains are identical could guide the future therapeutic approach. Nevertheless, we observed evolution in minority variants between BAL and NPS in patients #4 and #5, which was also suggested by at least two other studies [19,22], and could suggest some form of compartmentalization.

Evolution of minority variants points to the replicative nature of viral strains; this study thus supports that positive samples collected after 3 weeks from infection contain replicative viruses and not only residual genetic material.

Perez-Lago et al. described the minor variant landscape among three patients treated with anti-CD20 antibodies: most SNVs were minority/intermediate variants and were found outside the S gene [22]. Similarly, in our study, except for D614G, no mutation or minority variants were observed in the S gene coding for the spike protein; most were observed within the ORF1ab gene, which encodes the replication complex responsible for RNA synthesis. Whether the mutation observed could have contributed to the prolonged virus shedding in the studied cases remains unexplored. Siqueira et al. did not find any correlation between viral genetic diversity and clinical outcome but did observe a higher rate of viral genetic diversity among cancer patients compared to healthcare controls [18]. 

Compared to other groups, we did not detect a high number of SNVs. This could be explained by the short timing and limited number of samples per patient, and the absence of repetitive treatments that could have induced a selective pressure [3,22]. Perez-Lago et al. also reported a heterogenous viral diversity among three patients, with one patient presenting less variability.

There are some limitations in this study: Firstly, only a few patients had several samples usable for HTS analysis. Due to lower viral loads and, by extension, insufficient coverage, sequencing could not be used for samples collected after 3 weeks from diagnosis, precluding any conclusions for long-term shedders. Similarly, the lower viral loads in collected BAL and plasma samples did not allow us to have a representative selection of these specimens for sequencing data. A higher number of patients and samples and a longer follow-up period could have identified evolutionary convergence mutations [20]. Yet, although some case reports identified replicative virus up to 6 months [22] in specific situations and patients (especially with anti-CD20 treatment), this remains anecdotal, and virus shedding commonly does not exceed 3 weeks, as reflected by the CDC recommendations for transmission prevention [1]. Secondly, at the time of this study, different PCR methods were used daily in parallel for routine diagnostics. Quantitative values (IU/mL) were not available for each of the platforms used, and only Ct values obtained for each positive sample are reported. Thus, Ct values are provided here for informational purposes only and should be interpreted with caution. Finally, samples were collected in 2020, and circulating strains have since evolved, particularly with the rise in the Delta variant that is now predominant worldwide.

## 5. Conclusions

Our study reveals that among a cohort of oncological patients, there was no major and rapid evolution of viral strains in patients, both in samples collected 2–3 weeks apart and in samples collected in the upper and lower respiratory tracts. Nevertheless, we could identify minority variant evolution, reflecting a dynamic pattern of quasi-species replication; yet, no common pattern was observed among patients. Whether these findings have functional and clinical relevance is still unknown and needs further investigation. In the case of chronic replication, minority variants can become predominant and lead to evolutionary convergence and should be included in epidemiological surveillance.

## Figures and Tables

**Figure 1 microorganisms-09-02145-f001:**
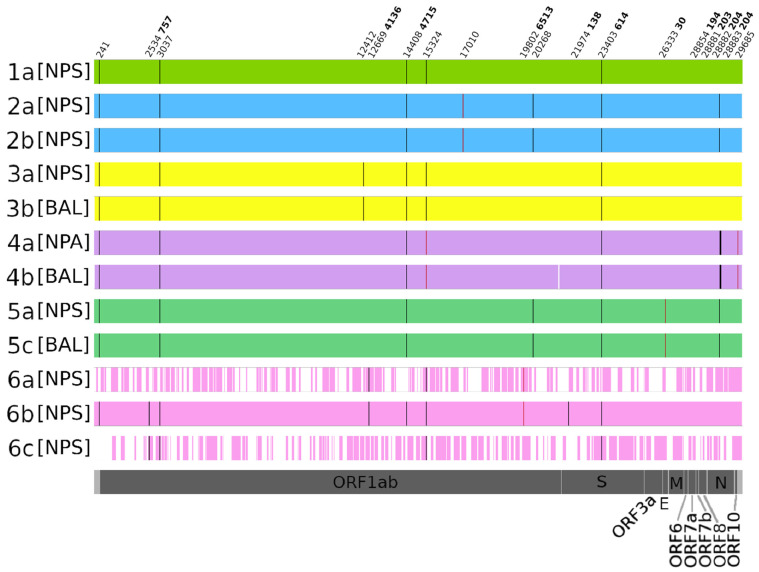
Sequencing results. Black bars represent positions with nucleotides that differ from the reference sequence MN908947. Red bars represent positions with minority nucleotide variants that differ from the reference sequence MN908947. White space represents regions whose coverage is lower than 10×. Nucleotide (normal font) and amino acid (bold font) changes are indicated at the top of the figure. Although the genome coverage obtained for sample 1b was too low to be reported in the figure, thus making the temporal variability analysis for this patient unfeasible, sample 1a is reported to show the presence of the nucleotide mutations at positions 241 and 3037 as well as D614G and P4715L in all samples. NPS = nasopharyngeal swab, NPA = nasopharyngeal aspiration, BAL = bronchoalveolar lavage, S = spike, E = envelope, M = matrix, N = nucleocapsid genes.

**Table 1 microorganisms-09-02145-t001:** SARS-CoV-2 RT-PCR assays of interest.

Nb of patients with ≥1 NPS	16
Median nb of NPS (range)	3 (2–9)
Nb of ≥1 positive NPS (%)	5 (31.2)
Median positivity duration, d (range)	28 (15–114)
Nb of patients with BAL	4
Median delay from NPS, d (range)	13 (7–22)
Nb positive (%)	4 (100)
Median CT value (range)	24 (19.8–25)
Nb of patients with viremia performed	21
Median delay from NPS, d (range)	9 (0–51)
Nb positive (%)	8 (38.1)
Median delay from NPS of positive samples, d (range)	8 (0–25)
Median CT value (range)	35.2 (32.4–39.3)

% proportion of the total analyzed for the same type of sample. NPS = nasopharyngeal swab, CT = cycle threshold, BAL = bronchoalveolar lavage, d = days.

**Table 2 microorganisms-09-02145-t002:** Characteristics of samples analyzed by HTS.

ID Patients	Sample	Sample Date	CT Value	Delay from First Sample (d)	Type of Investigation
1a	NPS	29.03.2020	22.2		intra-host temporal variability
1b	NPA	19.04.2020	26.22	21	
2a	NPS	27.03.2020	17		intra-host temporal variability
2b	NPS	15.04.2020	17.7	19	
2c	NPS	13.05.2020	31.7	47	
3a	NPS	13.03.2020	12.6		intra-host compartmental variability
3b	BAL	26.03.2020	19.8	13	
4a	NPA	01.04.2020	16.9		intra-host compartmental variability
4b	BAL	08.04.2020	23	7	
5a	NPS	31.03.2020	17		intra-host compartmental variability
5b	plasma	19.04.2020	30.6	19	
5c	BAL	22.04.2020	25	22	
5d	plasma	27.04.2020	30.5	27	
5e	BAL	03.05.2020	31	33	
6a	NPS	02.04.2020	26.3		intra-host temporal variability
6b	NPS	12.04.2020	20.7	10	
6c	NPS	24.04.2020	27	22	

Each ID patient number represents one distinct patient, the letter associated with the number represent distinct samples from the same patient. NPS = nasopharyngeal swab, NPA = nasopharyngeal aspiration, CT = cycle threshold, BAL = bronchoalveolar lavage, d = days.

## Data Availability

The raw sequence data were deposited in the NCBI Sequence Read Archive under BioProject accession number PRJNA755148.

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
