# Peer review of "SARS-CoV-2 Evolution among Oncological Population: In-Depth Virological Analysis of a Clinical Cohort"

_microorganisms, 2021, doi:10.3390/microorganisms9102145_

Round 1

Reviewer 1 Report

No changes required

Author Response

We thank the reviewer for this favorable opinion.

Reviewer 2 Report

This is an excellent study. I have only a minor issue; please check accuracy on lines 153-155 on page 5.

Author Response

We thank the reviewer for this faborable opinion and for his careful reading of the manuscript. Yes, the sentence is accurate, actually, in this patient, the wild-type strain was a minority variant in the first two samples before becoming the consensus strain in the last one.

Reviewer 3 Report

  1. Only 54 patients were involved in this study and their average age is 67.2 in this study. Please explain how to prove the result is significant because the sample size is so small and their age is too old.
  2. Please indicate the threshold value of CT in the RT-PCR assay. Additionally, please provide the viral load for each CT value using a X-Y curve.

Author Response

We thank the reviewer for his pertinent comments.

1. Our cohort of oncological patients has an average old age due to their oncological co-morbidity and we agree that results can thus not be extrapolated outside from oncological patients. This was specified in the first sentence of the discussion, p.6 : « The present study describes the clinical characteristics and outcomes of a single-center cohort of oncological patients, mostly with solid tumors, and results can thus not be extrapolated to other populations.»

According to the small sample size comment, this is true that only 54 patients were included in the cohort, of whom only 6 had sequencing data for identifying mutations. There were no more patients included in our cohort of oncological patients during the first wave in Geneva, because oncological patients were warned by their physicians to lockdown as much as possible so the incidence rate of infection was quite low among this population.

According to the retrospective nature of the study, we could only identify 6 patients and 12 samples suitable for sequencing ; we think our results are « significant » in the sense that mutations or minority variants were identified using a strong methodology and are thus objective results; nevertheless, we highlighted in the revised version the limitation of our results by commenting that time period between samples could have been too short to identify significant mutation (p. 7 : last paragraph of the discussion « Due to lower viral loads and, by extension, insufficient coverage, sequencing could not be used for samples collected after 3 weeks from diagnosis, precluding any conclusions for long-term shedders. … A higher number of patients and samples and longer follow-up period could have identified evolutionary convergence mutations»).

Finally, as described throughout the discussion section, our results are in line with other reports. As such, we think that our results represent a further step in understanding the genetic evolution of sars-cov-2, by confirming slow evolutionary rate but existence of underlying quasi-species.

2. As required by the reviewer, it is now mentioned in the revised version that for or all PCR assays, the positivity threshold was Ct values ⩽40 for at least one target (Chapter 2.2 “….was used on RNA previously extracted using the NucliSens easyMAG extraction kit bioMérieux). For all PCR assays, the positivity threshold was Ct values 40 for at least one target“).

In March-April 2020, our routine laboratory used every day different real-time RT-PCR methods in parallel to guarantee results in the shortest possible time and to anticipate potential supply disruptions. Therefore, the PCR results obtained in this study were generated by four different PCR methods (see methodology section). It was only much later that an international standard (i.e. WHO International Standard for SARS-CoV-2 RNA in IU /mL) was made available in order to harmonize the quantitative results (i.e viral load) between the different platforms / laboratories. Thus, only Ct values can be reported in this study (in order only to give a general overview). We agree that PCR Ct value should be interpreted with caution and represents a limitation of our study. This point is now mentioned in the revised version (p. 7 : last paragraph of the discussion Secondly, at the time of the study, different PCR methods were daily used in parallel for routine diagnostics. Quantitative values ​​(IU /mL) were not available for each of the platforms used and only Ct values ​​obtained for each positive sample are reported. Thus, Ct values are provided here for informational purposes only and should be interpreted with caution.”)

Reviewer 4 Report

Dear authors

The authors submitted an original manuscript entitled, "SARS-Cov-2 evolution among oncological population: in-depth virological analysis of a clinical cohort." The authors observed a cohort of patients with oncological diseases and SARS-CoV-2 infection and investigated viral evolution among selected patients. The study provides insights of SARS-CoV-2 infection and evolution in relatively immunocompromised patients by sequencing viral nucleic acids. The results were comprehensive and well-demonstrated. However, there are major issues needed to be clarified. The following are my comments.

  1. Since all patients enrolled were with oncological diseases, there was no control group to compare the difference of delayed viral shedding and frequent viral evolution between two groups. This is the major limitation and bias for the study to suppose that the immunocompromised patients may have delayed viral shedding and frequencies of viral mutation. Please adjust your study design or describe this issue in your discussion.
  2. Although there were 54 patients with an oncological disease in this cohort, only six patients’ samples were investigated by high-throughput sequencing. Conceptually, the non-nasopharyngeal samples may have a relatively low viral load, while a delayed samples may bear lower viral loads than the first sample. The authors should discuss the limitations in the limited cases analyzed with HTS, the difference of sample type, and the timing bias. All these factors may contribute to the molecular results and your conclusion.
  3. In the page 2 of materials and methods, the authors wrote, “HTS analysis was performed using the RNA protocol adapted from [13].” The protocol was adapted from a reference/ an article/ a company, but not “[13]”. Please describe it well.

Author Response

We thank the reviewer for his pertinent comments.

1. Contrary to Siqueira et al. (Ref 18 in the manuscript), we did not include a control group to compare the evolutionary rate of SARS-CoV-2 between « healthy » and immunocompromised hosts ». We did not aim at proving that oncological population has delayed viral shedding and frequencies of viral mutation, instead, we aimed at describing a cohort of oncological population and using their routinely collected samples to identify some viral evolution. Accordingly, we stated p. 6, 1st paragraph of the discussion « Of note, prolonged shedding has been described even in immunocompetent patients with viable virus that can be seen in samples analyzed up to 100 days after symptom onset » and completed in the revised version p.6-7: « As mentioned earlier, prolonged shedding has been described even in immunocompetent patients, and similarly, it is also possible that our findings of minority variants is not specific to oncological population, but could also be found in “healthy” infected individuals. »

2. We totally agree with the reviewer’s comment. Lower viral loads were indeed identified in NPS samples collected later during the infection and only 3 BAL samples were suitable for NGS analysis. We did not use plasma samples for NGS analysis. As stated above, we commented in the discussion section that we could not exclude that the absence of significant mutation observed is due to samples collected within a short period of time elapsed from infection (see question 1 Reviewer 3); in the revised version, we precised that this was due to low viral loads and added the same limitation for BAL and plasma sample : « Due to lower viral loads and, by extension, insufficient coverage, sequencing could not be used for samples collected after 3 weeks from diagnosis, precluding any conclusions for long-term shedders. Similarly, lower viral loads in collected BAL and plasma samples did not allow us to have a representative selection of these specimens for sequencing data. »

3. According to the reviewer remark, the sentence has been changed in the revised version to provide more clarity concerning the origin of the RNA protocol that we previously published Chapter 2.2HTS analysis was performed using the RNA protocol adapted from the study previously published by Petty et al. [13]. Briefly, for each specimen…”

Round 2

Reviewer 4 Report

Dear editor in chief and editorial team

The authors submitted a revised manuscript entitled, "SARS-Cov-2 evolution among oncological population: in-depth virological analysis of a clinical cohort." The authors observed a cohort of patients with oncological diseases and SARS-CoV-2 infection and investigated viral evolution among selected patients. The study provides real-world insights of SARS-CoV-2 infection and evolution in relatively immunocompromised patients by sequencing viral nucleic acids. The results were comprehensive and well-demonstrated. And the responses to reviewers are acceptable and appropriate. Acceptance is recommended.